# 3D Printing for Cartilage Replacement: A Preliminary Study to Explore New Polymers

**DOI:** 10.3390/polym14051044

**Published:** 2022-03-05

**Authors:** Gonçalo F. Delgado, Ana C. Pinho, Ana P. Piedade

**Affiliations:** Department of Mechanical Engineering, CEMMPRE, University of Coimbra, 3030-788 Coimbra, Portugal; goncalofldelgado98@gmail.com (G.F.D.); acdspinho@uc.pt (A.C.P.)

**Keywords:** cartilage tissue, 3D printing, Nylon^®^ 12, LAY-FOMM^®^ 60, mechanical properties

## Abstract

The use of additive manufacturing technologies for biomedical applications must begin with the knowledge of the material to be used, by envisaging a very specific application rather than a more general aim. In this work, the preliminary study was focused on considering the cartilaginous tissue. This biological tissue exhibits different characteristics, such as thickness and mechanical properties, depending on its specific function in the body. Due to the lack of vascularization, cartilage is a supporting connective tissue with limited capacity for recovery and regeneration. For this reason, any approach, whether to repair/regenerate or as a total replacement, needs to fulfill the adequate mechanical and chemical properties of the surrounding native cartilage to be successful. This work aims to explore the possibility of using new polymers for cartilage total replacement approaches with polymeric materials processed with the specific 3D printing technique of fused filament fabrication (FFF). The materials studied were Nylon^®^ 12 (PA12), already described for this purpose, and LAY-FOMM^®^ 60 (FOMM). FOMM has not been described in the literature for biomedical purposes. Therefore, the chemical, thermal, swelling capacity, and mechanical properties of the filaments were thoroughly characterized to better understand the structure–properties–application relationships of this new polymer. In addition, as the FFF technology is temperature based, the properties were also evaluated in the printed specimens. Due to the envisaged application, the specimens were also characterized in the wet state. When comparing the obtained results with the properties of native cartilage, it was possible to conclude that: (i) PA12 exhibits low swelling capacity, while FOMM, in its dry and wet forms, has a higher swelling capacity, closer to that of native cartilage; (ii) the mechanical properties of the polymeric materials, especially PA12, are higher than those of native cartilage; and (iii) from the mechanical properties evaluated by ultra-micro hardness tests, the values for FOMM indicate that this material could be a good alternative for cartilage replacement in older patients. This preliminary study, essentially devoted to expanding the frontiers of the current state of the art of new polymeric materials, provides valuable indications for future work targeting the envisaged applications.

## 1. Introduction

Cartilaginous tissue, or simply cartilage, is a supporting connective tissue composed of collagen, proteoglycan-rich matrix, and a single cell type, chondrocytes. This tissue differs from other human tissues due to its unique properties, especially a lack of blood vessels and nerve cells [1]. In the human body, cartilage formation is based on a process called chondrogenesis that, unfortunately, fails to enable the tissue to naturally self-repair after injury or degeneration [2].

Damage in cartilage can be induced by trauma and some clinical pathologies such as osteoarthritis [3]. Although the degeneration of cartilage is more common in the elderly due to sports activity, younger patients are increasingly being diagnosed [3]. Since this condition strongly interferes with the patients’ quality of life, an effective solution for the repair/replacement of cartilaginous tissue is needed.

As mentioned before, cartilage plays an important role in the human body, especially concerning supporting functions. Therefore, this tissue can adapt and bear mechanical loading while being able to deform and recover to the original volume, similar to a sponge with water [4]. Such demanding requirements increase the difficulty when designing devices for its substitution.

Nowadays, the two most common approaches for cartilage replacement and repair are total replacement, usually with cobalt–chrome (CoCr) or ultrahigh molecular weight polyethylene (UHMWPE)-based structures, and the scaffold implantation, in the case of tissue regeneration approach [5,6,7]. In the case of total replacement, the major challenges are related to the materials used and their mechanical performance, namely tensile strength and Young modulus. Even though CoCr is considered as a biocompatible and non-degradable material, its stiffness leads to the mechanical shielding of the bone induced by mechanical loading [7]. Although UHMWPE presents mechanical properties similar to those of native cartilage, it can be structurally unstable under loading, hindering its mechanical and tribological performance [6]. Moreover, due to wear, metal ions can be released, leading to genotoxic effects such as carcinogenicity and DNA damage [8,9,10].

For repair or regeneration purposes, biomedical scaffolds are often indicated as an advantageous solution as they can provide a 3D framework to enable cell proliferation, matrix deposition, and consequent tissue regeneration [11]. From work published in this area, the most common natural materials used in such devices are collagen, agarose, chitosan, hyaluronic acid, fibrin, and alginate [12,13,14]. Of the synthetic polymers, poly(ethylene glycol) (PEG) [15,16], poly(lactic acid) (PLA) [17,18], poly(vinyl alcohol) (PVA) [19,20] and polyurethane (PU) [21,22] are the most described and the produce best outputs.

For the design and architecture of the scaffolds and biomedical devices already reported for cartilage regeneration, structures such as membranes, hydrogels, and nanofibers produced by phase inversion [23], solvent-casting particle leaching [24], freeze–drying [25], and electrospinning [26] are among the approaches providing the most promising results. Nonetheless, the outcomes are still insufficient.

Additive manufacturing (AM) or 3D printing is a technology that enables the preparation of fully customizable scaffolds [27,28] with intricate shapes that can be designed using computer-aided design (CAD) or computed tomography (CT) data [29,30,31]. Due to the ease of processing and geometric freedom, this has been investigated for cartilaginous tissue regeneration purposes. Indeed, She et al. published a work describing the preparation of a scaffold prepared by 3D printing using two different materials [32]. The outside was a printed PCL hollow ring with a collagen sponge inside to mimic the anatomy of the native trachea of white rabbits. In vitro tests proved the growth of tracheal cartilage within the scaffold. The production of a silk fibroin-based scaffold with a 3D printed PCL mesh filling was also reported elsewhere [33]. 3D printed PCL/graphene composite scaffolds were also reported showing improved lubricity and drug-releasing properties [34].

In a different approach, a scaffold combining 3D printed polycarbonate–urethane (PCU) and UHMWPE was studied for the purposes of native lubrication mechanisms [35]. Unfortunately, surface roughness and consequent high friction coefficient have jeopardized its performance. In order to investigate the role of inner scaffold architecture, Jung et al. developed a 3D-printed PU tracheal scaffold with microscale design, which proved to be beneficial for cell infiltration and biological integration of the device [36]. Despite the number of publications and studies, each material and design approach needs to be directed to a specific type of cartilage and local implantation, which impairs the agreement and standardization of which route (material/processing technique) to follow.

PA12 is a semicrystalline polymer with excellent impact resistance at low temperatures, low water absorption, resistance to stress cracking, and fatigue under high-frequency cyclical loading conditions, which is frequently used in AM due to the feasibility of the process [37,38]. In addition, it is commonly used for applications related to the biomedical field, including for cartilage [39]. Therefore, it was used as a control material.

FOMM is a new commercially available material, and constitutes a mixture of two polymers, one of which is PVA. FOMM becomes flexible and porous when immersed in water due to the removal, by dissolution, of the PVA content. This characteristic may be of the main interest when applying this polymeric material for cartilage replacement. As cartilage does not contain blood vessels or nerves, and is supplied with nutrients through the compression and flexion of the tissue, it needs to have a porous structure to allow these interactions. Pitaru et al. published a work in which they use FOMM in an attempt to match the mechanical properties of native anterior cruciate ligaments, with promising results [40]. Nevertheless, the cited article is the only one concerning the consulted bibliography that presents research with FOMM material.

The present work describes a preliminary study exploring the possibility of using PA12 and FOMM for the preparation of structures, by 3D printing, for cartilage repair. To the best of our knowledge, this is the first time that such materials have been proposed for this specific application.

## 2. Materials and Methods

### 2.1. Materials

In the present work, polymeric filaments with a diameter of 1.75 ± 0.03 mm were used. Nylon^®^ 12 (PA12) filament was supplied by DoWire^®^ (Seixal, Portugal) and LAY-FOMM^®^ 60 (FOMM) filament was acquired from Filament2print^®^ (Nigrán, Spain). For comparative purposes, and prior to some characterization techniques, PA12 and FOMM filaments and printed parts were immersed in deionized water for four days (wPA12 and wFOMM, respectively).

### 2.2. Processing by 3D Printing

All specimens were printed using a FlashForge^TM^ Creator 3 3D printer (Ílhavo, Portugal) with a dual extruder, each with a 0.4 mm diameter nozzle. PA12 filament was printed at 260 °C with a bed temperature of 110 °C, while FOMM was printed at 230 °C with a bed temperature of 70 °C (Figure 1). The printing parameters for FOMM were previously optimized by varying a set of parameters that included: printing temperature from 220–250 °C, bed temperature from 30–80 °C, and printing speed from 15 to 30 mm·s^−1^. Both materials were printed at the same speed, 25 mm·s^−1^, with a 50% hexagonal infill pattern and 180 μm layer thickness. Two bottom and upper layers (100% infill with a linear pattern) were used to support and facilitate the specimen printing. The geometry of the printed specimens was chosen according to the requirements of the characterization technique, as discussed in the following sections.

### 2.3. Characterization

#### 2.3.1. Chemical Characterization

The infrared (IR) spectra of the studied filaments were acquired with FTNIR/MIR equipment (PerkinElmer, Frontier model, Waltham, MA, USA), equipped with an attenuated total reflectance (ATR), an FR-DTGS detector, and a KBr beam splitter, at 20 °C. For the data acquisition, the resolution was 4 cm^−1^, a constant force of 80 N, and 16 accumulation interferograms. PerkinElmer also supplied the ATR module with a diamond/ZnSe crystal. After the data collection, the spectrums were analyzed through the SPECTRUM 10 STD software.

#### 2.3.2. Thermal Characterization

The thermal stability of filaments and printed parts was studied using a TGA Q500 V20.13 equipment by TA instruments (New Castle, DE, USA), with a heating rate of 10 °C·min^−1^, between 25–600 °C, with a nitrogen flux of 50 mL·min^−1^. The results were analyzed using the TA Instruments Universal Analysis 2000 software supplied by the manufacturer.

The thermal events of the studied filaments and printed specimens were assessed using a DSC Q100 V9.9 equipment by TA instruments, with a heating rate of 10 °C·min^−1^ and a 50 mL·min^−1^ constant flux of nitrogen. The analysis of the results of the first heating cycle and the determination of the crystallization and enthalpies ($\Delta H_{\mathrm{cc}}$ and $\Delta H_{m}$, respectively) were performed using TA Instruments Universal Analysis 2000 software, supplied by TA Instruments. The percentage of crystallinity ($X_{c}$) was calculated using Equation (1):(1)$$X_{c}\;\left(\%\right)=\frac{\Delta H_{m}-\Delta H_{\mathrm{cc}}}{\Delta H_{\infty }}\times 100$$
where $\Delta H_{\infty }$ is a characteristic value of each material, corresponding to the melting enthalpy variation considering 100% of crystallinity [41].

The weight of the samples used for both thermal characterization techniques was kept constant at 8 mg.

#### 2.3.3. Morphological Characterization

The scanning electron microscopy (SEM) technique was used to observe the morphological dissimilarities between FOMM and wFOMM. The equipment used for the filament characterization was a ZEISS^®^ Merlin 61–50 Microscope (Carl Zeiss, Oberkochen, Germany), Gemini 2, with an accelerating voltage of 2 kV. Using a sputtering technique, all samples were coated with a 3 nm layer of gold. Samples were coated for 60 s with the help of EDWARDS EXC 120 sputtering equipment (Crawley, UK), with a power source Huttinger PFG 1500 DC (Schwaig bei Nuremberg, Germany). The sputtering conditions were: power, 0.11 kW; voltage, 1000 V; current, 1.83 A. The surface and cross-section morphologies of the printed specimens were characterized using an FEI Quanta 400FEG ESEM (FEI, Hillsboro, OR, USA). For the cross-sectional observation, the samples were immersed for 90 s in liquid nitrogen. This allowed for a clean fracture of the samples by mechanical impact. The printed samples were observed without any metallic coating.

#### 2.3.4. Swelling Capacity

The water uptake of the filaments studied at the present work was assessed by swelling capacity tests (SC). Five test samples of PA12, FOMM, and wFOMM filaments were dried at 50 °C until weight equilibrium and their initial weight collected. Then, all samples were immersed in 15 mL of ionized water at room temperature for seven days. The weight of the samples was collected every 24 h or 48 h and the water was substituted. The SC of the materials was determined through Equation (2):(2)$$\mathrm{SC}\;\left(\%\right)=\frac{W_{S}-W_{0}}{W_{S}}\times 100$$
where $W_{S}$ represents the swollen weight and $W_{0}$ is the initial dried weight [42].

#### 2.3.5. Mechanical Characterization

The tensile strength of the studied materials was determined using a Shimadzu apparatus, more specifically the Autograph AGS-X model (Tokyo, Japan), with a 5 kN load cell and a grip speed of 5 mm·min^−1^. All materials (PA12, wPA12, FOMM, and wFOMM) were tested at both filament (100 mm segments) and printed specimen (100 mm × 20 mm × 2 mm, according to ASTM D3039) configurations. Figure 2 shows a representative filament test.

Five samples of each material and form were considered for the study. For all tested materials, the distance between opposite ends, span, was 50 mm, and the obtained results were analyzed on Trapezium X software (Tokyo, Japan). The results were displayed in stress–strain curves, from which the calculation of Young’s modulus (E) was performed, according to Equation (3) [43],
(3)$$E=\frac{\sigma }{\varepsilon }$$
where *σ* refers to stress and ε is the strain.

Three-point bending (3PB) tests determined the flexural strength of the printed specimens. Five samples of each printed material were considered for the calculations. The dimensions of the testing specimens (60 mm × 10 mm × 2 mm) were chosen according to the ASTM Standard D790 recommendations. Tests were conducted using an Autograph AGS-X equipment from Shimadzu, with a 5 kN load cell and a displacement rate of 2 mm·min^−1^. The flexural strength ($\sigma_{f}$) was determined as the nominal stress in the middle span section obtained using the maximum load value, according to Equation (4),
(4)$$\sigma_{f}=\frac{3\mathrm{PL}}{2{\mathrm{bh}}^{2}}$$
where P refers to the maximum load, h and *b* are the thickness and the width of the specimen, respectively, and L represents the span length, which was kept constant at 40 mm. Flexural modulus ($E_{f}$) was determined following the linear elastic bending beams theory relationship, which can be expressed by Equation (5),
(5)$$E_{f}=\frac{\Delta {\mathrm{PL}}^{3}}{48\Delta \mathrm{uI}}$$
where ΔP is the load range, Δµ is the flexural displacement range, and I refers to the moment of inertia. $E_{f}$ was acquired by linear regression of the obtained load–displacement curves contemplating the interval in the linear segment with a correlation factor greater than 95%.

Ultra-microhardness characterization results were recorded by Fischerscope H100 equipment (Sindelfingen, Germany). Three printed specimens of PA12, wPA12, FOMM, and wFOMM were submitted to 5 indentation runs in 2 different areas. The test cycles consisted of a load–hold–unload function. The load rate was tuned so that each run would last about 60 s, with 30 s hold period at the maximum load for thermal drift correction. Six indentations were performed, in each run, at maximum load using 0.525 root increments, and 19 measurements in 30 s were made to access the creep value. The load increased from 0.4 mN to 1000 mN.

## 3. Results and Discussion

### 3.1. Filament Characterization

#### 3.1.1. Chemical Composition

The polymeric filaments were used as received. As is usual, suppliers do not share factual information on several aspects, including the percentage and type of additives mixed in the main polymeric material. For this reason, the chemical composition of PA12 and FOMM filaments was evaluated by FTIR. Poly(vinyl alcohol) (PVA) and wFOMM filaments were also analyzed and compared with the original FOMM spectrum to confirm the existence of PVA in the original formulation and its dissolution by immersion in water, as stated by the supplier. Figure 3 displays the obtained spectrum for each tested filament.

Figure 3a confirms the chemical structure of PA12, as it displays similarity with other spectra already reported in the literature [44]. The presence of the stretching vibration of N–H, CH_2_, and C=O at 3286 cm^−1^ (a), 3000–2800 cm^−1^ (b), and 1633 cm^−1^ (c), respectively, are highlighted; the overlapping of the bands corresponding to the bending vibration of C=O and the stretching vibration of C–N at 1537 cm^−1^ (d); and finally, the bending vibration of CH_2_ at 1447 cm^−1^ (e) [44]. Therefore, if any additives have been added to PA12, they are present in a residual concentration that will not affect the chemical properties of the polyamide.

For the analysis of FOMM results, it must be reminded that the literature lacks information concerning this material’s chemical composition. In addition, the supplier only refers to the presence of PVA and does not provide any more details concerning the other polymer. For this reason, a PVA filament spectrum was overlapped with FOMM to identify the peaks referring to PVA. From the comparison the spectra of PVA and FOMM, it is possible to identify the well-defined PVA peaks located between 3500–3000 cm^−1^ (f) related to the stretching vibrations of the O-H group and the stretching vibration of the C=O group between 1750–1650 cm^−1^ (c), even though they are slightly shifted. These variations were already expected since the mixture of PVA with another polymeric material, as reported by Alireza Kharazmi et al. obtained a similar outcome when ZnS nanoparticles were incorporated into PVA [45].

To identify the remaining FOMM peaks, Pitaru et al. proposed that FOMM is composed of a mixture of PVA and flexible thermoplastic polyurethane (TPU) [40]. This is the only published work that analyzes the other polymer present besides PVA, to the best of our knowledge. For this reason, the obtained wFOMM spectrum (with no PVA due to dissolution in water) was compared with TPU spectra from the literature. It is possible to observe the typical bands associated with TPU, such as the stretching vibration of the N–H group at 3350–3250 cm^−1^ (a), the band corresponding to CH_2_ between 2950–2850 cm^−1^ (b), the stretching vibration of C=O at 1750–1650 cm^−1^ (c), and the stretching vibration of C–N between 1260–1230 cm^−1^ (g). Since only the N–H stretching band is not common to PVA, it is not possible to fully conclude, at this stage, that TPU may be the other polymer mixed with PVA. However, considering the literature and the obtained results, this is a strong possibility.

#### 3.1.2. Thermal Characterization

Thermogravimetric analysis (TGA) was used to assess the thermal stability of the PA12 and FOMM filaments. The resulting thermogravimetric curves are plotted in Figure 4.

The thermal stability of materials is especially important when processing by 3D printing since it is a temperature-based process. For this reason, it is important to ensure that materials are extruded without jeopardizing their integrity. From the observation of Figure 4a it is possible to conclude that the decomposition of PA12 occurred within a single step between 375 °C and 500 °C, which was an expected result and in agreement with other results [46].

In the case of FOMM, once again, no direct comparisons can be established with the scientific literature due to the lack of studies of this polymer. Nonetheless, the obtained thermogravimetric curves exhibited three weight loss stages: around 100 °C, between 250–360 °C, and 360–475 °C. The first stage (100 °C) is assigned to the loss of water. In turn, the second and third stages (250–340 °C and 340–450 °C) match the decomposition stages of urethane bonds and polyol chains, respectively, and are usually found in TPU decomposition profiles [47]. One can then assume that TPU seems to be one of the counterparts that constitute the FOMM filament. However, one cannot exclude the information concerning the chemical composition of FOMM provided by the supplier which indicates that PVA is part of the composition of FOMM. For this reason, the profile obtained for FOMM was compared with pure PVA decomposition profiles found in the literature. Herein, three decomposition stages were found, and their temperatures also matched with FOMM. However, in the case of PVA, the second stage refers to the decomposition of bound water, which is water that is directly bonded to the polymeric structure, not only absorbed on the surface, and the third stage is assigned to the decomposition and consequent carbonization of the PVA network [48].

Since TPU and PVA degradation stages overlap and match the FOMM profile, the presence of TPU in the composition of FOMM could not be wholly confirmed by TGA measurements.

The onset and peak temperatures determined from the analysis of the displayed thermograms are presented in Table 1.

By comparing the values of the two filaments, it is evident that PA12 has superior thermal stability and can withstand temperatures close to 400 °C, as all decomposition and onset temperatures are above this value. The experimentally determined T_on_ of PA12 was 433.1 °C; above this temperature, PA12 starts to disintegrate and does not maintain its structural integrity. On the other hand, the T_on_ of FOMM occurs slightly before the material loses 5% of its mass.

The thermal events of the filaments were studied by DSC to correctly define the printing parameters according to the thermal transitions of the materials. The resulting curves are plotted in Figure 5 and the determined transition temperatures are presented in Table 2.

The heat flux curve of PA12 presents three different thermal events at specific temperatures, where the first endothermic reaction, at 107.6 °C, corresponds to the glass transition temperature (T_g_). Then, at 143.6 °C, the plot displays an exothermic curve which indicates the cold crystallization (T_cc_) of the polymeric structure. Finally, at 246.6 °C, the material undergoes melting (T_m_). The determined values of $\Delta H_{\mathrm{cc}}$ and $\Delta H_{m}$ were 3.25 J·g^−1^ and 19.06 J·g^−1^, respectively. Assuming that PA12 $\Delta H_{\infty }$ is 209.3 J·g^−1^ [41], the calculated value for $X_{c}$ was 6.7%.

Regarding FOMM, the identification of the thermal events was first established considering the known data available in the literature for the same PVA sample used in the previous FTIR analysis. Pure PVA presents a single T_g_ close to around 80 °C [49], which was also observed in the FOMM curve, and thus reinforced the possible presence of PVA in FOMM.

Then, the FOMM profile was compared with heat flux curves of TPU available in the literature to confirm if TPU was part of FOMM composition. As reported elsewhere [31], TPU displays two glass transitions, the first negative and the second around 70–80 °C. In addition, TPU has a melting temperature, T_m_, between 150–160 °C [50], also observed in the FOMM profile. Since these three transitions can be observed in FOMM, it can be concluded that FOMM contains PVA and TPU. DSC measurements were crucial for the selection of printing temperatures since they should be higher than the T_m_ of the materials (246.6 °C for PA12 and 155.9 °C for FOMM).

#### 3.1.3. Morphological Characterization

The morphology of FOMM and wFOMM filaments was observed by SEM, and the obtained micrographs are displayed in Figure 6.

From the SEM micrographs, it is possible to observe that the structure of wFOMM is more porous when compared with the as-received material. The absence of PVA can explain such a fact due to its dissolution in deionized water. Consequently, the space previously occupied by PVA was left empty, creating pores in the FOMM structure. Thus, SEM micrographs confirm the successful removal of PVA from the FOMM structure, which had already been hypothesized in the discussion of the thermal characterization results.

#### 3.1.4. Swelling Capacity

In the presence of water, native cartilage presents a sponge-like behavior as it increases its total volume due to water uptake. Nonetheless, it can release the water into the medium when mechanically deformed to maintain the system equilibrium. Despite this fact, its permeability is extremely low [51]. Swelling capacity (SC) tests were conducted to assess the behavior and affinity of the studied filaments with water. The results are plotted in Figure 7.

The profile exhibited by PA12 is characteristic of a hydrophobic material displaying only 3% of SC. On the other hand, FOMM and wFOMM present swelling capacity percentages of ca. 72.6% and 79.6%, respectively. Such difference can be related to the increased amount of porous and consequent free space in the structure of wFOMM, which favors water penetration into the polymeric network [52].

The content of PVA in the original FOMM network was calculated using SC measurements by the subtraction of the FOMM initial and final weights, equivalten to the dissolution of PVA during the test. The PVA content in FOMM initial structure represented 15% of the total weight.

The SC measured from adult meniscus cartilage is approximately 70% [53]. Therefore, considering the SC properties, FOMM and wFOMM seem to be more suitable for replacing such tissue. Therefore, a multimaterial approach using PA12 and FOMM may be interesting since the SC of the combined structure would decrease compared to FOMM alone, which may be beneficial for reducing tissue disturbance and interaction with the surrounding media.

After the printing step, the swelling capacity of the specimens was evaluated. As expected, the results did not present any significant difference with those of the filaments, since the processing technique does not induce any chemical modification and, therefore, the interaction of the materials with water was not affected.

#### 3.1.5. Tensile Tests

Tensile strength tests determined the mechanical behavior of the filaments. Furthermore, dry and wet samples were tested since the envisaged application of the studied materials concerns the in vivo environment. The wet samples were immersed in water until maximum swelling was reached. All tests were conducted until fracture except for FOMM-based materials, which were stopped at 20 N of maximum load due to their ductile behavior. The maximum load (P), σ, and ε parameters determined are summarized in Table 3.

From the PA12 and wPA12 results, it can be concluded that the wetting process contributes to the decrease in the mechanical resistance of the filament. Indeed, it is possible to observe a 24% decrease in P and σ parameters and a 15% reduction in the value of ε. Such behavior may be related to the plasticizer effect of water [33], even though PA12 showed low water absorption, as discussed earlier. Nonetheless, at the surface level, the water present in the material (that was not dried after immersion) slightly reduced the tensile strength and strain at break of the PA12 filament.

In the case of FOMM and wFOMM, it is noteworthy to mention that none of the tensile tests conducted were performed until failure due to the materials’ flexibility. The FOMM filament proved to be more resistant than wFOMM owing to its more compact structure [54] and PVA content [55], in contrast to the porous structure of wFOMM created by the diffusion of PVA.

### 3.2. Characterization of Printed Specimens

#### 3.2.1. Differential Scanning Calorimetry

The FFF technique is temperature based, where polymeric materials are extruded layer by layer to form 3D objects with pre-defined geometry. It is important to ensure that the polymeric materials do not suffer degradation during the extrusion step. In the present work, this was evaluated by TGA measurements, and the printing temperature adjusted accordingly. Nonetheless, the temperature cycle to which the materials were subjected may cause some changes in the polymeric thermal events. For this reason, DSC measurements were conducted in printed specimens.

When comparing the DSC PA12 curves obtained prior and after printing, it can be noticed that the peak profile is the same but with slight peak deviations. Such facts may be related to the alignment of the polymeric chains induced by processing, namely the extrusion process. It is reported in the literature that when materials are extruded, their polymeric chains tend to align and turn into a more organized network [56]. These results are supported by the literature where it is well established that the level of organization of a polymeric network influences the temperature at which the thermal events occur, as more organized structures require a higher amount of energy to trigger the movement of the chains [57].

Regarding FOMM (Figure 8), a slight transition temperature deviation was also noticed in the heat flux curves for the printed samples with and without PVA (Table 4). These deviations might be explained by the alignment of the polymeric chains induced by extrusion process and the reorganization of the polymeric network after the dissolution of PVA. Furthermore, the T_g_ at approximately 80 °C disappeared in wFOMM which means that PVA was completely removed. However, for both PA12 and FOMM, the temperature deviations are not significant for the envisaged application.

#### 3.2.2. Tensile Tests

The tensile strength tests were repeated for dry and wet printed specimens (wFOMM not dried) to investigate if the processing technique influenced the mechanical performance of the used materials. Table 5 summarizes the main values obtained by this technique.

No significant differences were observed in the values obtained for dry and wet PA12 samples. However, it is known that the printing process contributes to the alignment of the polymeric chains [58], which may jeopardize the penetration of water into the polymeric network and thus avoid water interference.

Commercial filaments are processed by extrusion in one direction, which is always the same as the pulling force of the tensile strength apparatus favoring mechanical performance. PA-printed specimens can sustain higher loads, but the tensile strength and strain at break were lower than those observed for the filaments. On the other hand, printed specimens are constructed layer by layer and, in the present work, with 50% of infill. Considering that the printing parameters and process influence the mechanical properties of materials [59], the empty spaces inside the specimen structures and the cohesion between layers may have contributed to the decrease in PA12 performance.

In the case of FOMM, following the same behavior already observed for filaments, wFOMM, owing to its less compact structure as confirmed by SEM observation of the cross-sections (Figure 9), displays lower mechanical resistance than the printed specimens with the original FOMM. When comparing FOMM filaments and printed specimens, similar behavior to PA12 was observed. Since printed specimens were designed with 50% infill, their structures are by default less compact than the original filament. Therefore, filaments are more resistant than printed specimens. These results reinforce the role of the printing parameters in the mechanical performance of the materials.

#### 3.2.3. Flexural Tests

In an attempt to simulate the loading supported by cartilage tissue, three-point bending tests were conducted to determine the $\sigma_{f}$ and $E_{f}$ of dry and wet printed specimens (Figure 10).

The data from Figure 10 imply that the results displayed by PA12 and wPA12 are quite similar, showing that water does not affect the flexural behavior of the PA12-based specimens. On the other hand, the FOMM and wFOMM stress–displacement profiles are quite different as the wFOMM structure highly deforms under lower loading values. The calculated $\sigma_{f}$ and $E_{f}$ values of the printed specimens are presented in Figure 11.

The calculated values for PA12 agree with the literature [31]. In the comparison between PA12 and wPA12, a slight decrease in the flexural modulus (ca. 5%) can be observed, while the maximum flexural strength increases by around 3% for the wet specimens. Such behavior can be explained by the plasticizer effect of water [33], which directly influences the flexural modulus as the material becomes slightly less stiff.

In the case of FOMM, considering the assumption that it is mostly composed of TPU, it was expected to display an elastomer-like behavior. Indeed, both FOMM and wFOMM showed similar profiles to those found for TPU [50], with low $E_{f}$ and $\sigma_{f}$ values. However, the flexural strength of wFOMM is significantly lower than FOMM due to the porous structure of wFOMM. The original structure of FOMM already behaves like an elastomer, and is therefore quite deformable. If the structure presents porosity, and consequently free spaces, it will deform even more under the same load. For this reason, both $\sigma_{f}$ and $E_{f}$ values will significantly decrease, as proven by the experiments.

For cartilage replacement, materials should match the mechanical properties of native cartilage to reduce possible implant malfunction and rejection and avoid the need for revision surgery [60]. However, cartilage presents distinct mechanical properties according to the different functions of this tissue in different parts of the human body. For this reason, when considering materials for cartilage repair, it is important to focus on a specific type of tissue. In the literature, studies provide some information on the mechanical properties of different types of cartilage, such as septum and rib cartilage, whose $\sigma_{f}$ and $E_{f}$ values are reported to be approximately 1.5 MPa and 7.2 MPa for the septum and 24.3 MPa and 8.8 MPa for the rib, respectively (Table 6) [60].

A comparison between our results and the literature reveals some promising similarities. wFOMM has an $E_{f}$ of 14.4 ± 2.0 MPa, which is relatively close that of rib cartilage and septum cartilage. In turn, FOMM (210.4 ± 39.0 MPa), wPA12 (1005.0 ± 44.5 MPa), and PA12 (1059.4 ± 49.6 MPa) all have higher E_f_ values compared with native cartilage. Nonetheless, 3PB cartilage values available in the literature [61] show that deeper layers of cartilage tissue, such as the zone calcified cartilage (ZCC), which serves as a transitional zone between the deep zone of cartilage and subchondral bone, are closer to those obtained for dry and wet PA12 (around 1000 MPa). Considering the results from this work and the reported studies in the literature, the combination of FOMM and PA12 could be an attractive approach for cartilage replacement, with a possible structure constituted by an inner part of PA12 to provide strength and structure to the implant, surrounded by FOMM in the outer part to mimic the native cartilage performance.

#### 3.2.4. Ultra-Microhardness

Ultra-microhardness experiments were conducted to assess the reduced modulus ($E_{r})$ and Vickers hardness ($H_{v})$ values of the printed specimens for comparison of the results with the literature, as these tests are often reported for cartilage-related studies. The results are plotted in Figure 12.

The results show that both materials present higher $E_{r}$ values than those calculated by 3PB tests. Since ultra-microhardness only evaluates the properties of the outer layers (shells with 100% infill), contrary to what is verified for macroscale techniques such as 3PB, these results were expected [62].

wPA12 displays higher modulus and hardness values when compared with the dry specimens, in contrast to what was observed in the macroscale tests. This can be related to the influence of water at the molecular level. The chemical structure of PA12 comprises N–H bonds that can be protonated by water, forming NH_3_^+^ [62]. As a result of these interactions, at the molecular scale, instead of plasticizing the material, water will have a deleterious effect on the mobility of the polymeric chains by stiffening the structure due to a pseudo-crosslinking effect.

In the case of wFOMM, the porosity created by the absence of PVA does not seem to affect the $E_{r}$ and $H_{v}$ of the material negatively. Once again, as the depth range of the ultra-microhardness tests does not comprise the inner porosity of wFOMM, this explains the calculated values. Considering the envisaged application, Table 6 summarizes data collected from the literature concerning the mechanical properties of native cartilage, with different functions and determined by different characterization techniques.

Compared to the $H_{v}$ values of native cartilage obtained by μ-hardness (0.5 MPa for septum and 1 MPa for rib) [60], those of PA12, in both the dry and wet forms, are notably higher. On the contrary, FOMM results are closer to the human rib cartilage evaluated by the same technique. However, it is important to note that the listed values from native cartilage refer to male patients between 63 and 86 years old [60]. Chondrocytes begin to dissipate from the superficial region with increased age, and accumulate in the deeper layers. As a result, the hydration decreases, and the matrix becomes stiffer [65]. Thus, these values should be lower in younger patients. Nevertheless, FOMM presents closer results to native cartilage than PA12, indicating that this material could be a good alternative for cartilage replacement in older patients. Since the prevalence of cartilage diseases is higher in older patients, the impact of FOMM results is even more relevant.

## 4. Conclusions

The present work aimed to produce 3D printed structures that could be used in the biomedical field, specifically for cartilage repair/substitution. This work also intended to expand the frontiers of knowledge using a polymeric material rarely reported in the literature, FOMM. The study also examined PA12. Both the dry and wet forms of each materials were subjected to filament characterization (chemical, thermal, and mechanical), confirming the chemical composition of PA12 and, in the case of FOMM, calculating the amount of PVA in its structure (15%). In addition, it was concluded that FOMM was composed of PVA and TPU.

3PB tests conducted on dry and wet specimens showed that wFOMM samples had compelling results similar to native cartilage. The SC of FOMM and wFOMM proved to be similar to native cartilage. PA12, in turn, exhibited a poor swelling rate, which could be helpful for cartilage repair/regeneration in a multimaterial approach.

In the ultra-microhardness test, as expected, all test pieces had higher $E_{r}$ compared with the 3PB test. However, wPA12 displayed a stiffer behavior than dry PA12 owing to the molecular interactions between water and N–H groups. On the other hand, FOMM and wFOMM presented similar results to native cartilage of older patients. This similarity could be beneficial as most cases of cartilage replacement occur in aged patients. The obtained results provide promising evidence that 3D printed parts may be part of the future of regenerative medicine. Furthermore, this study highlighted new research paths, such as designing multimaterial structures with a PA12 core and outer shell in wFOMM. Future work should include the preparation of such multimaterial structures and their in vitro characterization, which will include prokaryotic and eukaryotic cell tests.

## Figures and Tables

**Figure 1 polymers-14-01044-f001:**
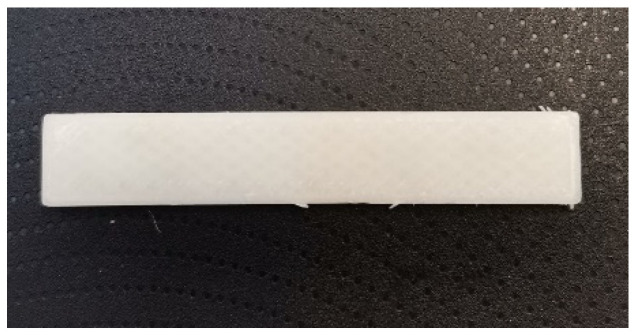
Macrograph of FOMM printed test specimen for flexural tests.

**Figure 2 polymers-14-01044-f002:**
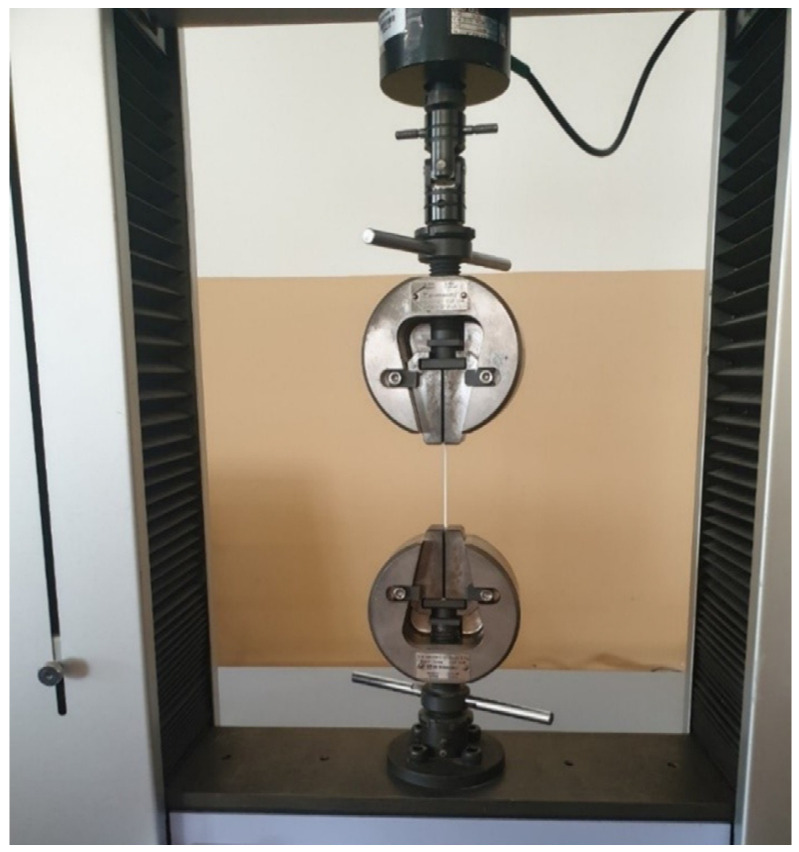
Representative macrograph of tensile test of the FOMM filaments.

**Figure 3 polymers-14-01044-f003:**
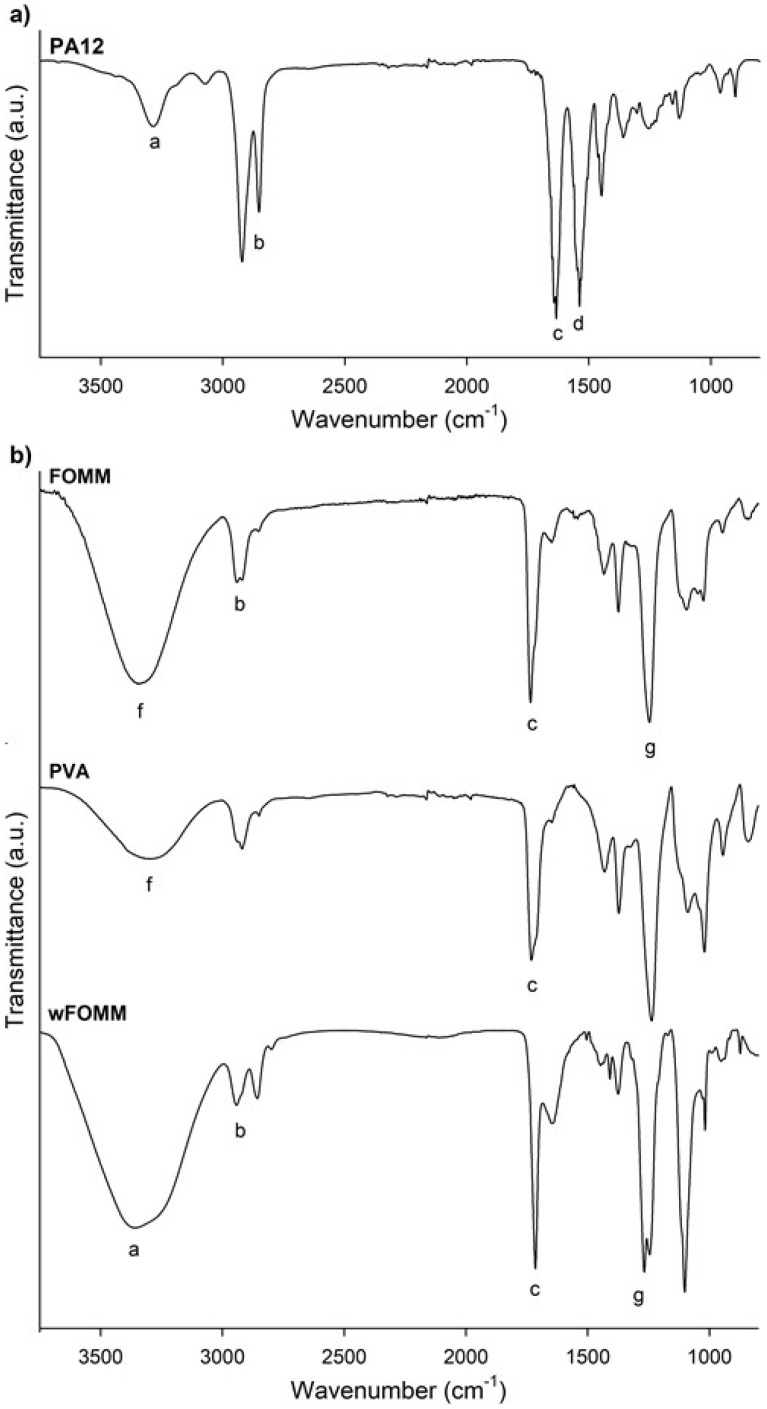
FTIR spectra of the polymeric filaments: (**a**) PA12, (**b**) FOMM, PVA and wFOMM. The letters (a–g) identify the characteristic bands discussed in the text.

**Figure 4 polymers-14-01044-f004:**
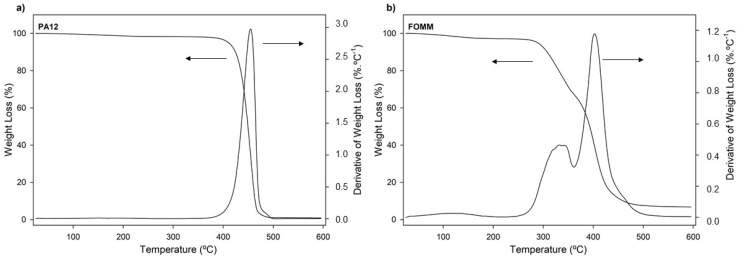
Weight loss and derivative of weight loss (DTG) thermogravimetric curves of (**a**) PA12 and (**b**) FOMM, as received.

**Figure 5 polymers-14-01044-f005:**
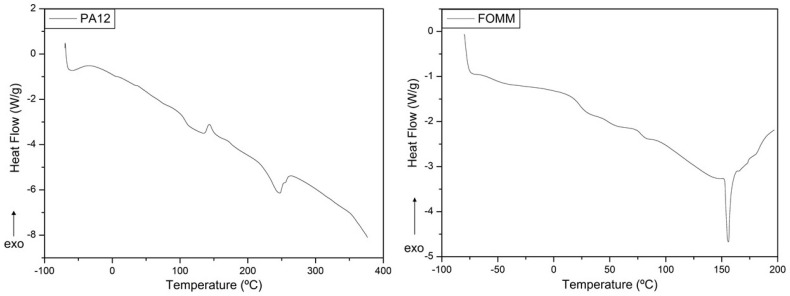
Heat flux curves obtained for PA12 and FOMM filaments, as received.

**Figure 6 polymers-14-01044-f006:**
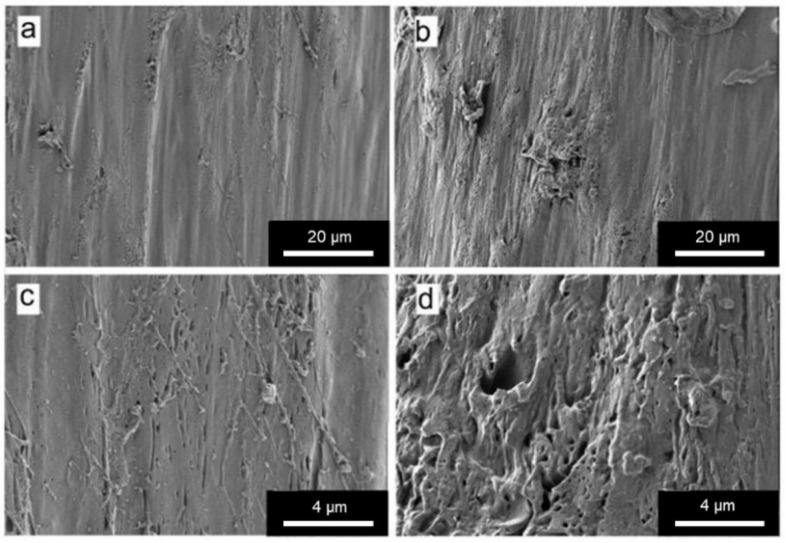
SEM micrographs of (**a**) and (**c**) FOMM filament; (**b**) and (**d**) wFOMM filament. Scale bars (**a**) and (**b**) = 20 µm; (**c**) and (**d**) = 4 µm.

**Figure 7 polymers-14-01044-f007:**
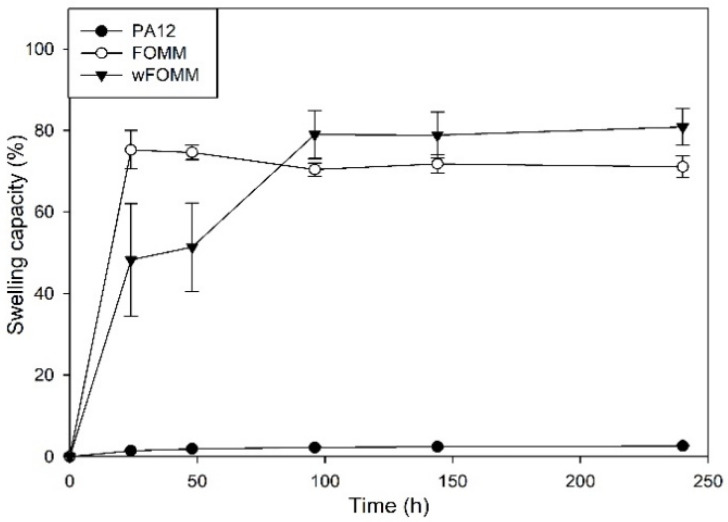
SC of the studied filaments. Results are presented as mean ± SD.

**Figure 8 polymers-14-01044-f008:**
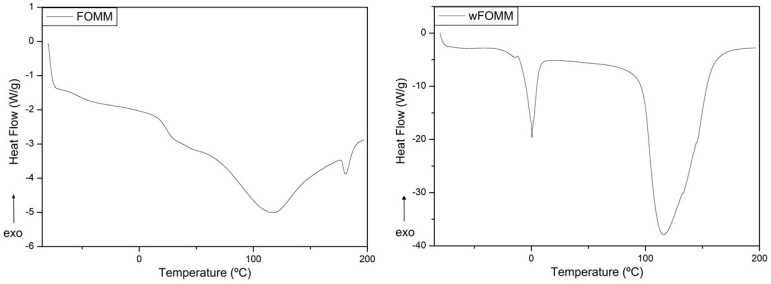
DSC of printed FOMM and wFOMM specimens.

**Figure 9 polymers-14-01044-f009:**
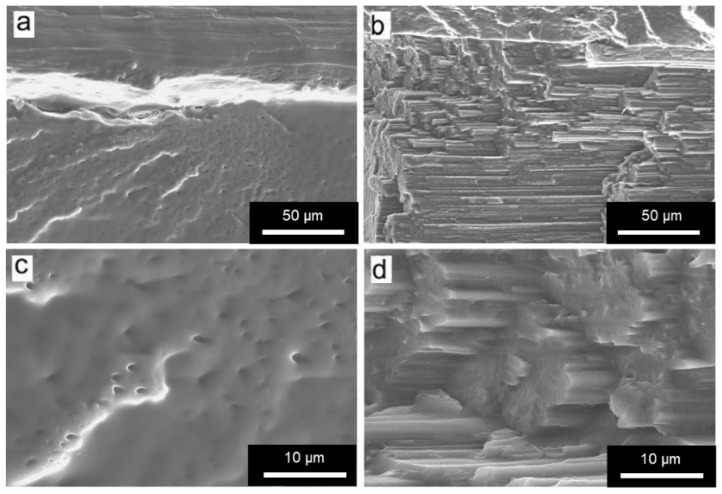
SEM micrographs of printed specimens of (**a**,**c**) FOMM t; (**b**,**d**) wFOMM. Scale bar (**a**,**b**) = 50 µm; (**c**,**d**) = 10 µm.

**Figure 10 polymers-14-01044-f010:**
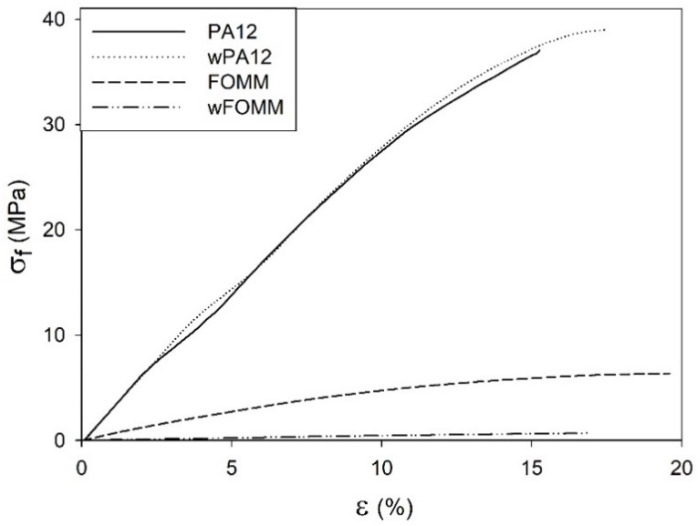
Stress–strain representative 3PB curves of printed specimens.

**Figure 11 polymers-14-01044-f011:**
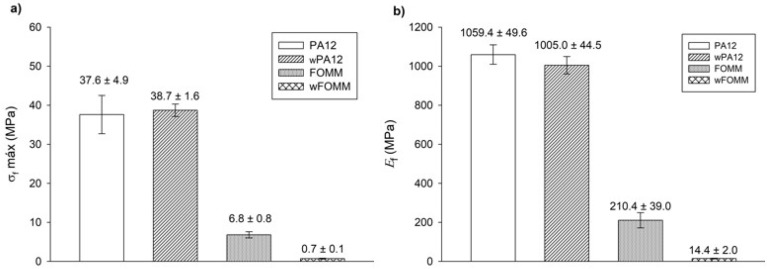
3PB test: (**a**) maximum flexural strength and (**b**) flexural modulus. Results presented as mean ± SD.

**Figure 12 polymers-14-01044-f012:**
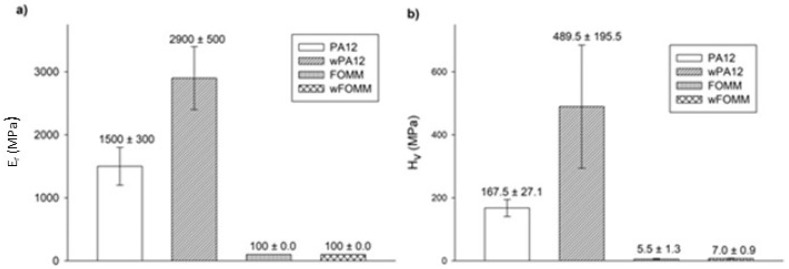
$E_{r}$ (**a**) and $H_{v}$ (**b**) determined by ultra-microhardness tests. Results presented as mean ± SD.

**Table 1 polymers-14-01044-t001:** Reference temperatures obtained by TGA.

Filament	T_on_ (°C)	T_5%_ (°C)	T_10%_ (°C)	T_p1_ (°C)	T_p2_ (°C)
PA12	433.1	416.3	426.9	454.6	-
FOMM	296.5	298.7	314.4	333.5	403.1

T_on_—Onset temperature; T_5%_—Temperature to which corresponds 5% of weight loss; T_10%_—Temperature to which corresponds 10% of weight loss; T_p_—peak temperature.

**Table 2 polymers-14-01044-t002:** Transition temperatures obtained by DSC of the filaments.

Filament	T_g1_ (°C)	T_g2_ (°C)	T_cc_ (°C)	T_m_ (°C)
PA12	107.6	-	143.6	246.6
FOMM	−42.9	82.4	-	155.9

**Table 3 polymers-14-01044-t003:** Tensile strength results of PA12, wPA12, FOMM, and wFOMM filaments.

	PA12	wPA12	FOMM	wFOMM
P (N)	110.2 ± 9.4	83.3 ± 13.4	20.0 ± 0.0	6.0 ± 0.2
σ (MPa)	45.8 ± 3.9	34.6 ± 5.6	8.3 ± 0.0 *	2.5 ± 0.0 *
ε (%)	14.6 ± 9.9	12.4 ± 3.6	1.4 ± 0.0 *	5.0 ± 0.2 *

* value at 20 N of applied load.

**Table 4 polymers-14-01044-t004:** Transition temperatures of the FOMM printed specimens obtained by DSC.

Filament	T_g1_ (°C)	T_g2_ (°C)	T_m_ (°C)
FOMM	−51.6	79.6	117.3 to 181.4
wFOMM	−24.9 to 0.7	-	116.3

**Table 5 polymers-14-01044-t005:** Tensile strength results obtained for dry and wet printed specimens.

	PA12	wPA12	FOMM	wFOMM
P (N)	951.7 ± 103.7	920.3 ± 157.9	252.3 ± 14.2	55.6 ± 4.9
σ (MPa)	23.8 ± 2.6	23.0 ± 4.0	6.3 ± 0.4 *	1.4 ± 0.1 *
ε (%)	11.4 ± 6.2	11.4 ± 8.2	12.6 ± 0.7 *	2.8 ± 0.2 *

* value at 20 N of load applied.

**Table 6 polymers-14-01044-t006:** Mechanical properties of native human cartilage.

Cartilage	σ_f_ (MPa)	E_f_ (MPa)	E (MPa	H_v_ (MPa)	Technique	Ref.
Septum	-	-	951.7	0.50	μ-hardness	[60]
Rib	-	-	23.8	1	μ-hardness	[60]
Rib	24.3	8.8	-	-	Tensile tests	[60]
Septum	1.5	7.2	-	-	Tensile tests	[60]
Knee	-	-	5.8	-	μ-hardness	[63]
Articular cartilage	-	-	4.4	0.31	μ-hardness	[64]

## Data Availability

The data presented in this study are available on request from the corresponding author.

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
