# Peer review of "3D Printing for Cartilage Replacement: A Preliminary Study to Explore New Polymers"

_polymers, 2022, doi:10.3390/polym14051044_

Round 1

Reviewer 1 Report

The scientific paper "3D printing for cartilage replacement: A preliminary study to explore new polymers” aimed to explore the possibility of using new polymers for cartilage total replacement approaches with polymeric materials processed with the specific 3D printing technique of fused filament fabrication (FFF). It can be considered that:

From the texts marked in yellow, the manuscript must have already passed through other reviewers (perhaps it is a resubmission). The manuscript presents a detailed methodology of the experimental protocol, with consistent and clear results and discussion. If the authors consider it relevant, they could analyze the inclusion or not of a reference in the area (Osseointegration Improvement of Co-Cr-Mo Alloy Produced by Additive Manufacturing) published in the journal Pharmaceutics MDPI in 2021. Were there any limitations in the study? If yes, include at the end of the discussion or in the conclusions. It should also indicate the next step of the experiment, that is, an in vivo study.

Author Response

The scientific paper "3D printing for cartilage replacement: A preliminary study to explore new polymers” aimed to explore the possibility of using new polymers for cartilage total replacement approaches with polymeric materials processed with the specific 3D printing technique of fused filament fabrication (FFF). It can be considered that:

From the texts marked in yellow, the manuscript must have already passed through other reviewers (perhaps it is a resubmission). The manuscript presents a detailed methodology of the experimental protocol, with consistent and clear results and discussion. If the authors consider it relevant, they could analyze the inclusion or not of a reference in the area (Osseointegration Improvement of Co-Cr-Mo Alloy Produced by Additive Manufacturing) published in the journal Pharmaceutics MDPI in 2021. Were there any limitations in the study? If yes, include at the end of the discussion or in the conclusions. It should also indicate the next step of the experiment, that is, an in vivo study.

Answer: The authors acknowledge the comments from the reviewers. Regarding the inclusion of the reference “Osseointegration Improvement of Co-Cr-Mo Alloy Produced by Additive Manufacturing”, the authors did not followed the suggestion, since it refers to a metallic alloy, and not a polymeric material. For this reason, the experiment design and conclusions are, as expected, different. The present manuscript describes a preliminary study, where the main object was to extensively characterize the POLYMERS under study, especially the FOMM. For this reason, no defined limitations were found. Regarding the next steps of the work, the authors added a paragraph in the conclusions section addressing this question. The authors would like to thank the reviewer for the time spent in the appreciation of the manuscript.

Reviewer 2 Report

Comments:

  1. The authors mentioned “additive manufacturing (AM) or 3D printing is a technology that enables the preparation of fully customizable scaffolds…” It is recommended to include some recent relevant, highly-cited papers on 3D bioprinting of tissues and scaffolds.
    1. “Print me an organ! Why we are not there yet." Progress in Polymer Science97 (2019): 101145.
    2. "3D bioprinting of tissues and organs for regenerative medicine." Advanced drug delivery reviews132 (2018): 296-332.

  1. It is also recommended to mention about the different 3D printing techniques (based on ASTM standards) that are suitable for fabricating tissue-engineered scaffolds.

  1. Can the authors elaborate more on the material composition of LAY-FOMM? What is the other polymer beside PVA?

  1. Please highlight/circle the area of interest for Figure 3 – FTIR spectrum.

  1. The authors have conducted extensive mechanical characterization; however one key missing data is on the biocompatibility of this new material as a material for tissue engineering applications.
    1. Please provide some cell biocompatibility test to demonstrate the ability to grow cells on the scaffolds

Author Response

The authors mentioned “additive manufacturing (AM) or 3D printing is a technology that enables the preparation of fully customizable scaffolds…” It is recommended to include some recent relevant, highly-cited papers on 3D bioprinting of tissues and scaffolds.

“Print me an organ! Why we are not there yet." Progress in Polymer Science97 (2019): 101145.

"3D bioprinting of tissues and organs for regenerative medicine." Advanced drug delivery reviews132 (2018): 296-332.

Answer: The authors acknowledge the comment from the reviewer and followed the recommendation of adding the abovementioned references. The reference list and numeration were altered accordingly.

It is also recommended to mention about the different 3D printing techniques (based on ASTM standards) that are suitable for fabricating tissue-engineered scaffolds.

Answer: The authors acknowledge the comment from the reviewer. However, in the introduction section the authors already provide an overview of the published work on 3D printing concerning cartilaginous tissue where the 3D printing techniques are discussed. For this reason, in the authors opinion, since only two techniques (bioprinting and fused filament fabrication (FFF)) are often reported, and they both share the same work principle, it would be repetitive to specify them in the text.

Can the authors elaborate more on the material composition of LAY-FOMM? What is the other polymer beside PVA?

Answer: The authors state throughout the manuscript that LAY-FOMM is constituted by PVA and TPU, which is the flexible and non-water-soluble counterpart of the original material. This assumption was supported by characterization techniques FTIR, TGA and DSC. The conclusion that the major constituent of FOMM is TPU, was only done after all the characterization techniques, because with the results of only one of them it was not possible to unequivocally conclude what polymer it was.

 Please highlight/circle the area of interest for Figure 3 – FTIR spectrum.

Answer: In figure 3, the authors identified the peaks of interest with letters, which are then mentioned in the discussion section of FTIR characterization. To each letter is assigned one band of interest for the identification of the chemical bond to which it refers. This is a commonly used strategy for the presentation of FTIR results in the literature and for that reason, in the opinion of the authors, figure 3 should be kept as it is.

The authors have conducted extensive mechanical characterization; however one key missing data is on the biocompatibility of this new material as a material for tissue engineering applications.

Answer: The reviewer is absolutely correct. Nonetheless, the present manuscript intends to present a preliminary study on the possibility to fabricate scaffolds for cartilaginous tissue related applications using a NEW POLYMER, FOMM. For this reason, the authors focused their work on the extensive characterization of the material in order to enrich the available literature on the matter. Since the results proved to be promising, the next step of the work foresees the in vitro validation of the printed structures. The authors would like to stress that, their work conduct does not agree with statements such as “ good for biomedical applications”. As researchers that work in the field clearly know, it is impossible for a material to fulfil all the requirements for every biomedical application. In this sense, the authors always try to focus on one specific application that they consider to be relevant and agrees with the properties of the materials under study.

Please provide some cell biocompatibility test to demonstrate the ability to grow cells on the scaffolds

Answer: As stated earlier, the present manuscript refers to a preliminary study, with focus on the characterization of the materials, which is the scope of the Polymers journal. The future work, including in vitro validation and cell biocompatibility studies will definitely be submitted to more specialized journals on the matter, such as Biomaterials, Acta Biomaterialia or Advanced Healthcare Materials.

Reviewer 3 Report

The reviewed article presents a comparison of two polymer materials (one reference) as a substitute for cartilage material, with the possibility of obtaining implants by means of 3D printing. I find the very idea of ​​the research problem very interesting. However, the article requires some changes before being accepted for printing. I focused my attention especially on the research of thermal properties. I have a few remarks and comments to the presented research results:

  1. What was the mass of samples in DSC and TG tests? As it is known, in the case of polymeric materials, mass is of great importance in thermal studies, and the comparison of two materials should be performed with similar masses.
  2. I strongly disagree with the interpretation of the TG results presented on page 8. First, assigning a weight loss in the range of 250-340°C to a water loss is wrong. It is far too high a temperature for this phenomenon. In fact, Fig. 4b clearly shows that there is one more weight loss, not described by the authors, in the range of up to 200°C. It is this loss that is associated with the loss of water. The loss of mass at a temperature of 250-340°C. I would rather attribute to the degradation of TPU, which the authors write about later in the text.
  3. I also disagree with the statement that PA12 can withstand temperatures above 400°C. At this temperature it probably is no longer good for anything.
  4. On what basis do the authors claim that PA12 maintains its properties up to a weight loss of 10%? What are the properties? Mechanical? Thermal?
  5. From which heating cycle of the DSC study were the results analyzed? From the first or the second? It is very important because we analyze completely different properties depending on the heating cycle.
  6. The degree of crystallinity for PA12 is 7.5% from the given values ​​for Hm, Hc and H100%. What was the degree of crystallinity of the FOMM?
  7. I would also change the interpretation of DSC results, especially for FOMM. The Tg PVA value found by me is 80°C. If the Authors agree with me, it changes the interpretation of the DSC curve for FOMM a little.
  8. Page 12 - the DSC PA12 curve of the printed elements and its comparison with the DSC of the filament would be useful.
  9. If the authors write that the transformation temperatures of filaments and printed elements did not differ significantly, they should show it, for example, in the table. For me, when looking at the DSC curves of filaments and printed elements, the differences are very large.
  10. The differences in the DSC FOMM curves of the filaments and the elements printed with FOMM and wFOMM are also very large, while the authors completely ignored this fact.

Author Response

The reviewed article presents a comparison of two polymer materials (one reference) as a substitute for cartilage material, with the possibility of obtaining implants by means of 3D printing. I find the very idea of ​​the research problem very interesting. However, the article requires some changes before being accepted for printing. I focused my attention especially on the research of thermal properties. I have a few remarks and comments to the presented research results:

  1. What was the mass of samples in DSC and TG tests? As it is known, in the case of polymeric materials, mass is of great importance in thermal studies, and the comparison of two materials should be performed with similar masses.

Answer: The authors acknowledge the comment from the reviewer. They are aware of the importance of maintaining similar sample masses in thermal characterization such as DSC and TGA. In this work, a sample mass of 8 mg was kept constant throughout the essays for both materials. This information was added to the manuscript.

  1. I strongly disagree with the interpretation of the TG results presented on page 8. First, assigning a weight loss in the range of 250-340°C to a water loss is wrong. It is far too high a temperature for this phenomenon. In fact, Fig. 4b clearly shows that there is one more weight loss, not described by the authors, in the range of up to 200°C. It is this loss that is associated with the loss of water. The loss of mass at a temperature of 250-340°C. I would rather attribute to the degradation of TPU, which the authors write about later in the text.

Answer: The reviewer is correct regarding the assignment of the degradation peaks. Herein, the authors might not have be clear about the discussion of these results. The first stage of weight loss of the FOMM filament (around 100ºC) refers to physiosorbed water loss, as expected. Then, the second stage (250-340ºC), is assigned to the degradation of urethane bonds, which is the first indicator that FOMM may be composed of TPU. The last peak (350-500ºC) is reported as the degradation of polyol chains. Both urethane and polyol are characteristic of the TPU material. After these conclusions, the authors compared the obtained results with those found in literature for PVA, and the possibility of this material to be one of the counterparts of FOMM. As reported in the literature, the weight loss profile of PVA has three peaks: the first, around 100ºC refers to the loss of physiosorbed water at the surface of PVA; the second (200-340ºC) is assigned to the loss of bound water, which is water that is bounded to the polymeric matrix, and not simply adsorbed on the surface. The presence of such peak is common for materials with high affinity with water, which is the case of PVA, as reported in the literature. Finally, the last peak (340-450ºC) is the stage where PVA completely degrades, leading to its carbonization. The fact that the decomposition stages of PVA and TPU reported in the literature are located at the exact same range of those of FOMM demands further investigation on the chemical composition of FOMM. At this point of the investigation, FOMM can be constituted by both, or only by PVA or TPU polymeric materials. In order to facilitate the comprehension of this discussion segment, the authors altered the manuscript, accordingly, hoping that this question becomes clearer.

  1. I also disagree with the statement that PA12 can withstand temperatures above 400°C. At this temperature it probably is no longer good for anything.

Answer: The authors acknowledge the comment from the reviewer, but they do not completely agree with the presented argument. The determined onset temperature of PA12 is 433.1ºC, which agrees with the literature. According to the ASTM, the onset temperature is “the point in the TGA curve where a deflection is first observed from the established baseline prior to the thermal event” and is known as the temperature at which the material starts to lose its properties. For this reason, supported by the obtained experimental data, it is safe to assume that at 400ºC, PA12 still maintains its integrity, and therefore, is considered to be thermally stable. This does not imply that the polymer can be used at this temperature. Nonetheless, the sentence in the manuscript was modified to clarify this fact.

  1. On what basis do the authors claim that PA12 maintains its properties up to a weight loss of 10%? What are the properties? Mechanical? Thermal?

Answer: The answer to this question is related with the above. The onset temperature establishes the thermal stability since it is considered that above this temperature, the material starts to disintegrate and no longer maintains its structural integrity. Since the temperature at which PA12 loses 10% of its weight (426.9ºC) is under the onset temperature (433.1ºC), it can be assumed that even after losing around 10% of its weight, PA12 is able to main its molecular integrity. To facilitate the comprehension, the authors have changed the sentence accordingly.

  1. From which heating cycle of the DSC study were the results analyzed? From the first or the second? It is very important because we analyze completely different properties depending on the heating cycle.

Answer: The authors agree with the reviewer statement. In this work, only first DSC cycles were considered because the filament is going to be used as received and is not going through a “first heating cycle” prior to its processing. This information was included in the materials and methods section.

  1. The degree of crystallinity for PA12 is 7.5% from the given values ​​for Hm, Hc and H100%. What was the degree of crystallinity of the FOMM?

Answer: The degree of crystallinity for FOMM is not possible to calculate because there is no  value already established (corresponding to 100% crystallinity). Furthermore, assuming that FOMM is mainly constituted by TPU, which is a designation of a class of materials with variable chemical composition (since the suppliers do not reveal this information), and not a specific material as PA12 with clearly defined and known chemical composition, it is not possible to correctly determine  .

  1. I would also change the interpretation of DSC results, especially for FOMM. The Tg PVA value found by me is 80°C. If the Authors agree with me, it changes the interpretation of the DSC curve for FOMM a little.

Answer: The authors acknowledge the comment from the reviewer. Indeed, there was a mistake in the assignment of the Tg of PVA. The only thermal event assigned to PVA that can be identified in the FOMM profile it it’s Tg at 80ºC. Indeed, pure PVA only present this specific thermal event since it is amorphous. On the other hand, and considering the TPU thermal events found in the literature, the TPU profile matches the FOMM profile in three thermal events: a negative first Tg, a second Tg between 70-80ºC, and finally, a Tm between 150-160ºC. The overlapping of the thermal events of TPU with FOMM reinforces the idea that TPU is a counterpart of FOMM. The Tg of PVA is overlapped with the second Tg of TPU. The text was corrected accordingly.

  1. Page 12 - the DSC PA12 curve of the printed elements and its comparison with the DSC of the filament would be useful.

Answer: The authors acknowledge the comment from the reviewer. However, as stated in the manuscript, the DSC profile is quite similar prior and after printing, with a slight deviation in the temperature range, as explained in the manuscript. For that reason, the authors consider that the difference between curves is not significant enough to be included in the manuscript, contrary to what is observed for FOMM.

  1. If the authors write that the transformation temperatures of filaments and printed elements did not differ significantly, they should show it, for example, in the table. For me, when looking at the DSC curves of filaments and printed elements, the differences are very large.

Answer: The authors appreciate the comment from the reviewer. A table with the thermal events obtained by DSC of the FOMM printed specimens was included in the manuscript.

  1. The differences in the DSC FOMM curves of the filaments and the elements printed with FOMM and wFOMM are also very large, while the authors completely ignored this fact.

Answer: The authors disagree with the reviewer. Indeed, in line 425, a possible explanation is given. Since there is not much information on FOMM, the authors attributed the profile changes between filament and printed specimens to the extrusion process, which is associated with chain alignment, which influences the behavior of the materials. However, this assumption cannot be supported by the literature since no similar information was yet published.

Reviewer 4 Report

Dear,
In general, this work aims to explore the possibility of using new polymers for total cartilage replacement approaches with polymeric materials processed by the special 3D printing technique of Fused Filament Fabrication (FFF). The materials investigated are Nylon® 12 (PA12), which has already been described for this purpose, and LAY -FOMM® 60 (FOMM). The latter has never been described in the literature for biomedical purposes. Therefore, the chemical, thermal, swelling and mechanical properties of the filaments were thoroughly characterised to better understand the of this new polymer.
My concerns are the following:
1. To arouse the readers' interest, authors should provide more information on the mechanical properties of cartilage, especially on cartilage wear and shear behaviour in the human body. Furthermore, give more information related to mechanical propertiesof recently studied materials. This is important in order to relate the newly studied material to recently used materials and to cartilage, especially at this preliminary stage.
2. The authors should perform a comprehensive statistical analysis. This is important as the authors have entered a new field with this very specific material.
Stating a few statistical conclusions based on conducting experiments with five samples is not sufficient. Consider testing this material immersed in, e.g. in Ringer's solution.
3. The authors should thoroughly discuss the properties of this new material compared to the current state of the art. The focus should be on wear, shear and surface roughness. This is important in view of the future interaction with the human body (e.g. possible use in the human knee joint, etc.).

The work presented in this manuscript is very promising. Keep up the good work!
Kind regards!

Author Response

In general, this work aims to explore the possibility of using new polymers for total cartilage replacement approaches with polymeric materials processed by the special 3D printing technique of Fused Filament Fabrication (FFF). The materials investigated are Nylon® 12 (PA12), which has already been described for this purpose, and LAY -FOMM® 60 (FOMM). The latter has never been described in the literature for biomedical purposes. Therefore, the chemical, thermal, swelling and mechanical properties of the filaments were thoroughly characterised to better understand the of this new polymer.
My concerns are the following:
1. To arouse the readers' interest, authors should provide more information on the mechanical properties of cartilage, especially on cartilage wear and shear behaviour in the human body. Furthermore, give more information related to mechanical properties of recently studied materials. This is important in order to relate the newly studied material to recently used materials and to cartilage, especially at this preliminary stage.

Answer: The authors agree with the reviewer’s opinion. However, unfortunately, it is quite difficult to assess the mechanical properties of cartilaginous tissue in the literature. The authors found that, most of the times, researchers base their works on ultra-microhardness as first approach. For this reason, in order to facilitate the establishment of comparisons, the authors followed the same strategy. Nonetheless, the materials under study were submitted to tensile and flexural tests. The use and study of PA12, already reported for cartilage related applications, served the purpose of comparing FOMM to a previously reported material. The choice of PA12 was also due to the fact that it can be processed by 3D printing, and therefore, the two materials under study could be processed by the same technique.

  1. The authors should perform a comprehensive statistical analysis. This is important as the authors have entered a new field with this very specific material.
    Stating a few statistical conclusions based on conducting experiments with five samples is not sufficient. Consider testing this material immersed in, e.g. in Ringer's solution.

Answer: The reviewer is absolutely correct. However, as previously mentioned, this manuscript refers to a preliminary study, where the focus was to characterize FOMM material, and explore the possibility to use it for cartilaginous tissue related applications. Since the study was in its preliminary phase, no statistical data was conducted. Indeed, since the available literature on FOMM is lacking, the authors think that the literature would benefit more from a first glance at FOMM and its properties. The in vitro validation of the material, in turn, is the object of the next publication which will follow the present one, which will be more focused on the biologic profile of the material.

  1. The authors should thoroughly discuss the properties of this new material compared to the current state of the art. The focus should be on wear, shear and surface roughness. This is important in view of the future interaction with the human body (e.g. possible use in the human knee joint, etc.).

Answer: The authors acknowledge the comment from the reviewer. When designing the experiment, the authors have considered the most cited mechanical tests reported in the literature for cartilaginous tissue. Indeed, for soft applications, the most used is ultra-microhardness, which the authors included in the present manuscript. For this reason, wear, shear and surface roughness were not addressed, as it is not easily comparable with the available literature on the subject. However, the authors appreciate the suggestion and will consider such tests in their future work.

The work presented in this manuscript is very promising. Keep up the good work!
Kind regards!

Answer: Thank you very much for your encouraging words.

Round 2

Reviewer 2 Report

The revised manuscript can be accepted in present form. 

Author Response

The authors acknowledge the reviewer comments.

Best regards

Reviewer 3 Report

The authors fully answered the questions contained in the first review. The article may be accepted for publication.

Author Response

Dear reviewer,

Thank you very much for your appreciation and for the time spent with our manuscript

Reviewer 4 Report

Dear authors,
I think it's important to compare your results with the mechanical properties of cartilage. I understand that you used PA12 as a control material, but that is not enough to claim that your new material has chances in cartilage repair. 
Please read

  1. http://www.scielo.org.mx/scielo.php?script=sci_arttext&pid=S1665-73812014000100006
  2. https://www.ncbi.nlm.nih.gov/pmc/articles/PMC2175072/
  3. .http://web.mit.edu/cortiz/www/3.052/3.052CourseReader/27_BiomechanicsofCartilage.pdf

etc.

See these links for the mechanical properties of cartilage. The articles are available for free.
Make a table or chart so that readers can see some similarities or major differences in relation to your results. A good example is your reference #40 with ACL and its comparison with tested materials.

Discuss this and then possibly claim that your future work with cartilage has potential.

Kind regards!

Author Response

Please find the reply to the reviewer in the attached file

Round 3

Reviewer 4 Report

Dear,

The authors have improved their manuscript providing an adequate response.

I have no further complaints!

Kind regards! 

Author Response

Dear reviewer,

Thank you for your comment and for the time spent with our manuscript